# Switching Heavy Chain Constant Domains Denatures the Paratope 3D Architecture of Influenza Monoclonal Antibodies

**DOI:** 10.3390/pathogens12010051

**Published:** 2022-12-28

**Authors:** Moffat M. Malisheni, Cheng-Shoong Chong, Tanusya M. Murali, Kiren Purushotorman, Xinlei Qian, Alfred Laiman, Yee-Joo Tan, Paul A. MacAry

**Affiliations:** 1Institute of Molecular and Cell Biology, Agency for Science, Technology and Research (A*STAR), Singapore 138673, Singapore; 2Department of Microbiology and Immunology, Yong Loo Lin School of Medicine, National University of Singapore, Singapore 117545, Singapore; 3Immunology Programme, Yong Loo Lin School of Medicine, National University of Singapore, Singapore 117456, Singapore; 4Bioinformatics Institute, Agency for Science, Technology and Research (A*STAR), Singapore 138671, Singapore; 5Integrative Sciences and Engineering Programme, National University of Singapore Graduate School, National University of Singapore, Singapore 119077, Singapore; 6Infectious Diseases Translational Research Program, Yong Loo Lin School of Medicine, National University of Singapore, Singapore 117545, Singapore

**Keywords:** heavy chain constant domain, paratope, epitope, VR-analogous IgG variants, influenza, monoclonal Ab

## Abstract

Several human monoclonal Abs for treating Influenza have been evaluated in clinical trials with limited success despite demonstrating superiority in preclinical animal models including mice. To conduct efficacy studies in mice, human monoclonal Abs are genetically engineered to contain mouse heavy chain constant domain to facilitate the engagement of Fc-receptors on mouse immune effector cells. Although studies have consistently reported discrepancies in Ab effectiveness following genetic engineering, the structural and mechanistic basis for these inconsistencies remain uncharacterized. Here, we use homology modeling to predict variable region (VR) analogous monoclonal Abs possessing human IgG1, mouse IgG1, and mouse IgG2a heavy chain constant domains. We then examine predicted 3D structures for variations in the spatial location and orientation of corresponding paratope amino acid residues. By structurally aligning crystal structures of Fabs in complex with hemagglutinin (HA), we show that corresponding paratope amino acid residues for VR-analogous human IgG1, mouse IgG1, and mouse IgG2a monoclonal Abs interact differentially with HA suggesting that their epitopes might not be identical. To demonstrate that variations in the paratope 3D fine architecture have implications for Ab specificity and effectiveness, we genetically engineered VR-analogous human IgG1, human IgG4, mouse IgG1, and mouse IgG2a monoclonal Abs and explored their specificity and effectiveness in protecting MDCK cells from infection by pandemic H1N1 and H3N2 Influenza viruses. We found that VR-analogous monoclonal Abs placed on mouse heavy chain constant domains were more efficacious at protecting MDCK cells from Influenza virus infection relative to those on human heavy chain constant domains. Interestingly, mouse but not human heavy chain constant domains increased target breadth in some monoclonal Abs. These data suggest that heavy chain constant domain sequences play a role in shaping Ab repertoires that go beyond class or sub-class differences in immune effector recruitment. This represents a facet of Ab biology that can potentially be exploited to improve the scope and utilization of current therapeutic or prophylactic candidates for influenza.

## 1. Introduction

Globally, influenza viruses infect over 1 billion people every year. Five million of these infections result in severe cases with over 650,000 succumbing to Influenza [1]. The cost associated with Influenza in the United States alone exceeds 11 billion dollars [2] indicating that Influenza viruses pose a very serious threat to both public health and the economy warranting the urgent need to develop more effective tools for preventing and controlling Influenza. Hemagglutinin (HA), one of the major glycoproteins on the surface of Influenza viruses is responsible for attaching Influenza viruses to sialic acid receptors present on the surface of host cells via the receptor binding site (RBS) and for releasing viral RNA into the host cell cytoplasm through a pH mediated conformational change of the fusion peptide [3]. Hence, Abs that target the RBS or fusion peptide of HA can prevent Influenza viruses from attaching to and infecting host cells. The effectiveness of several anti-Influenza virus monoclonal Abs has been explored in clinical trials with disappointing outcomes because they failed to demonstrate a superior health benefit when compared to placebo or commercially available antiviral drugs [4]. This is despite demonstrating superiority in animal models like mice [5,6,7]. Human monoclonal Abs are often genetically engineered to possess mouse heavy chain constant domains (IgG1 and IgG2a) and the same human heavy chain variable regions (VRs), also termed VR-analogous monoclonal Abs, before being evaluated in mouse models [8,9,10]. Most studies use binding assays like the enzyme-linked immunosorbent assay (ELISA) to demonstrate that the binding profiles of VR-analogous monoclonal Abs are similar [9]. However, showing similar binding profiles does not always translate into similar neutralization profiles warranting the need to perform both assays. Extremely rare studies use both ELISA and neutralization assay to validate the consistency of binding and neutralization profiles for VR-analogous human and mouse IgG variants [8]. Although several studies have illustrated functional inconsistencies between VR-analogous monoclonal Abs [11,12], the structural and mechanistic basis for these discrepancies remains uncharacterized.

An Ab comprises four polypeptide chains. Two identical heavy and light polypeptide chains both consisting of variable and heavy chain constant domains. Variable domains of light and heavy polypeptide chains contain three complementarity-determining regions (CDRs) and four Framework regions. Together, these regions determine Ab specificity and effectiveness. A group of CDR and Framework amino acid residues that directly interact with a set of amino acids on the antigen is referred to as the paratope and epitope, respectively [13]. Heavy chain constant domains determine the class/subclass of Abs and are important for mediating immune effector functions by engaging Fc receptors on immune effector cells. IgG is the most abundant Ab class in blood and is further subdivided into IgG1–4. Mouse IgG1, IgG2a, and IgG2b are considered orthologs of human IgG subclasses [14].

Homology modeling can predict 3D structures comparable to X-ray crystallography provided the sequence similarity exceeds 50% [15,16]. This is because proteins with similar amino acid sequences have conserved 3D structures [16,17]. Below this threshold, conformational variations increase in magnitude with decreasing amino acid sequence similarity [15,18]. Here, we use homology modeling to predict 3D structures for VR-analogous human IgG1, mouse IgG1, and mouse IgG2a. To show how corresponding paratope amino acid residues for predicted 3D structures will interact with HA, 3D X-ray structures for anti-Influenza virus Fabs in complexes with HA were structurally aligned with predicted 3D structures. Furthermore, we genetically engineered, expressed, and purified VR-analogous human IgG1, mouse IgG1, and mouse IgG2a to explore the role of human and mouse heavy chain constant domain sequences in influencing the specificity and effectiveness of monoclonal Abs. We demonstrate that swapping human IgG1 and IgG4 heavy chain constant domains with mouse IgG1 or mouse IgG2a heavy chain constant domains repositions and reorients paratope amino acid residues consequently modifying their interactions with epitope amino acid residues on HA. These structural changes led to broadened Ab specificity and potent protection of MDCK cells from Influenza virus infection. These data might imply that the corresponding paratope amino acid residues for VR-analogous human IgG1, mouse IgG1, and mouse IgG2a could be interacting with different epitope amino acid residues and could have significant implications for monoclonal Ab or vaccine efficacy studies in mouse models.

## 2. Materials and Methods

### 2.1. Modeling and Analysis of 3D Structures

We randomly selected F045-092, CR6261, FI6, and 5J8 human monoclonal Abs that target influenza virus HA. F045-092 and 5J8 target the RBS, while CR6261 and FI6 bind to the fusion peptide on HA. Light and heavy chain variable domain sequences for F045-092, CR6261, FI6, and 5J8 were derived from the Protein Data Bank (PDB), while heavy chain constant domain sequences for human IgG1, mouse IgG1, and mouse IgG2a were identified from the IMGT Repertoire (https://www.imgt.org/IMGTrepertoire/Proteins/, accessed on 13 October 2022). Characteristics of all monoclonal Abs utilized in this study are summarized in Table 1, while their respective sequences are presented in the Appendix A.

Validation of predicted human IgG1 and mouse IgG1/IgG2a Abs.

Before structure prediction, alignment files (.pir files) were created by aligning the sequences of the light and heavy chains of F045-092, CR6261, FI6, and 5J8 (fasta files) with the corresponding ligand-unbound template chains, B12 (human IgG1), Mab-61.1.3 (mouse IgG1), and MAB231 (mouse IgG2a), using MAFFT v7.45 with the ‘auto’ and ‘reorder’ arguments. Where possible, missing amino acids in the template PDB files were fixed using the BuildSymRes function in Yasara Structure v18.2.7 [19] which fills in missing amino acids using symmetry-related information from members of an oligomer.

To predict the structures of model anti-Influenza IgG variants sharing analogous VR sequences but differing heavy chain constant domain sequences, homology modeling was performed with the help of a Modeller (v9.21) [20] script written in Python (v2.7) based on instructions found at multi-chain modeling (https://salilab.org/modeller/manual/node29.html, accessed on 13 October 2022). For each target Ab, 5 predicted structures were created and the best model out of the 5 based on the score from the MODELLER scoring function (molpdf score) was chosen for further structural analysis.

Analysis of 3D structures was performed using YASARA View (http://www.yasara.org/, accessed on 13 October 2022). Heavy chains of predicted human IgG1, mouse IgG1, and mouse IgG2a were structurally aligned with VR-analogous heavy chains of Fab-HA complex X-ray structures. Paratope amino acid residues were exposed to explore the influence of heavy chain constant domains on their spatial location and orientation.

We analyzed heavy chains because genetic alterations (mutations) only involved switching heavy chain constant domains. We presented structural changes happening in paratopes since paratope amino acid residues are solely responsible for interacting directly with epitope amino acid residues. Heavy chains were exclusively used for simplicity and clarity. All VR-analogous IgG variants used the same light chains to ensure that any differences in their paratope architecture can be attributed to variations in their heavy chain constant domains.

### 2.2. Determining Distances between Paratope and Epitope Amino Acid Residues

We used YASARA View to measure the distance between donor/acceptor atoms on paratope amino acid residues and acceptor/donor atoms on epitope amino acid residues. Hydrogen bond distance is presented in angstroms.

### 2.3. Cell Lines

Human embryonic kidney-293 (HEK-293) were cultured in FreeStyle F17 expression medium (Gibco, cat. # A13835-01) supplemented with 1% Pluronic F-68 (Gibco, cat. # 24040-032), 0.05% G418 (Geneticin, cat. # 10131035) and 2% L-glutamine (Gibco, cat. # 25030081) while Madin–Darby canine kidney (MDCK) cells were cultured in Dulbecco’s Modified Eagle Medium (DMEM) (Gibco, cat. # SH30243.01) supplemented with 10% fetal bovine serum (FBS; Gibco, cat. # SH30071.03). Both cell types were incubated at 37 °C with 5% CO_2_.

### 2.4. Cloning of Recombinant HA

A/New Caledonia/20/1999 (NC H1N1; Genbank accession # AAP34324.1), A/California/07/2009 (pH1N1) (Genbank accession # ACP41953.1), A/Aichi/2/68 (H3N2A; Genbank accession # BAN81712.1), and A/Perth/16/2009 (H3N2P; Genbank accession # ACS71642.1). HA sequences for NCH1N1, pH1N1, H3N2A, and H3N2P were individually cloned into pTT5 expression vectors obtained from the National Research Council of Canada (NRCC). The pTT5 plasmid map is presented in Appendix A.

### 2.5. Influenza Viruses

A/California/07/2009 (pH1N1) and A/Aichi/2/1968 (H3N2A) influenza viruses were obtained from Associate Professor Vincent Chow from the National University of Singapore, Department of Microbiology and Immunology.

### 2.6. Cloning of VR-Analogous IgG Variants

PTT5 expression plasmids were genetically engineered to contain either human IgG1 or mouse IgG1/IgG2a heavy chain constant domains. Heavy and light chain VR sequences for F045-092, CR6261, FI6, and 5J8 were submitted to Twist Bioscience for gene synthesis. VR sequences for F045-092, CR6261, FI6, and 5J8 were digested from cloning vectors and ligated into pTT5 expression plasmids containing either human IgG1/IgG4/IgG1 with a mouse IgG1 hinge or mouse IgG1/IgG2a/IgA heavy chain constant domains. Corresponding full light chains were equally cloned into respective pTT5 expression plasmids.

### 2.7. Expression of HA

Briefly, HEK-293 cells were transfected with HA-expressing pTT5 plasmids preincubated with polyethylenimine (PEI) and maintained as mentioned above. Triptone-N1 was added twenty-four hours post-transfection to improve the expression efficiency. HA proteins were purified from cell culture supernatants using His Trap columns from GE Healthcare, now Cytiva. The purity of HA proteins was visualized using Coomassie blue staining.

### 2.8. Expression of VR-Analogous Ab Variants

Briefly, HEK-293 cells were transfected simultaneously with respective pairs of heavy and light chain pTT5 expression vectors, preincubated with polyethylenimine (PEI), and maintained as mentioned above. Triptone-N1 was added twenty-four hours post-transfection to improve the expression efficiency. VR-analogous Abs were purified from cell culture supernatants using Protein G for mouse IgG variants and MabSelect for human IgG subclasses. The purity of HA proteins was visualized using Coomassie blue staining.

### 2.9. Immunofluorescence Assay (IFA)

MDCK cells were trypsinized (0.125% trypsin-EDTA) and resuspended in DMEM containing 10% FBS. After cell density and viability determination, approximately 100,000 MDCK cells were seeded on 12 mm coverslips placed in a 24-well plate and incubated at 37 °C overnight. The cells were then maintained in serum-free DMEM for at least two hours prior to infection with influenza viruses (0.1 MOI). Infected MDCK cells were incubated overnight at 37 °C. MDCK cells were then fixed with 4% PFA for 30 min, after which cells were blocked with 2% FBS in 0.5% PBST for an hour. Cells were then incubated with F045-092, CR6261, FI6, and 5J8 human IgG1 for an hour and then washed three times with PBS before incubating with mouse anti-human IgG1 Fc (Alexa Fluor^®^ 488) for an hour followed by washing as above. MDCK cells were counterstained with 4′,6-diamidino-2-phenylindole (DAPI; Sigma-Aldrich, St. Louis, MO, USA) for 5 min, after which coverslips containing cells were mounted on microscope glass slides and analyzed by confocal microscopy.

### 2.10. ELISA

Ninety-six well plates (Thermo Fisher Scientific, Waltham, MA, USA; cat. # 442404) were coated with recombinant HA proteins from various influenza virus strains diluted in PBS and incubated overnight at 4 °C. An equal amount of bovine serum albumin (BSA; Thermo Fisher Scientific, cat. # 23209) was coated to serve as a negative control. The plates were washed 3 times with 0.05% Tween-20, followed by PBS before blocking with 5% FBS in 0.05% PBST (blocking buffer) for an hour at room temperature. After washing, purified human and mouse Abs diluted in blocking buffer were added to the plates and incubated for 1–2 h at room temperature. Following washing, horseradish peroxidase (HRP)-conjugated goat anti-human (Thermo Fisher Scientific, cat. # 31413) or anti-mouse (Thermo Fisher Scientific, cat. # 31439) Abs diluted in blocking buffer were added to plates and incubated for an hour at room temperature. The plates were washed and developed using 3,3′,5,5′ tetramethylbenzidine (TMB; Thermo Fisher Scientific cat. # 34029). The TMB–HRP reaction was stopped with 2M H_2_SO_4_ (Sigma Aldrich, cat. # 7664-93-9). Absorbance readings (OD_450_ nm) were determined using a plate reader (BioTek Instruments, Inc, Winooski, VT, USA).

### 2.11. Neutralization Assay

The median tissue culture infectious dose (TCID_50_) of each influenza virus strain was determined using WHO guidelines [21] MDCK cells (30,000–40,000) per well were seeded in 96-well plates (Thermo Scientific, cat. # 167008) and incubated overnight. Equal amounts of serially diluted anti-HA human and mouse Abs beginning with 200 µ/mL were added to equal amounts of influenza virus particles. Ab-virus mixtures were incubated for 2 h at 37 °C, after which they were added to MDCK cells and incubated for an hour at 37 °C. Two hundred microliters of virus growth medium (486.5 mL DMEM, 13.5 mL Hepes (Lonza, Basel, Switzerland, cat. # 17-737E) and 0.5 µg/mL TPCK-trypsin [1:4000 dilution of stock 2 mg/mL]) were added to the wells, and the plates were incubated at 37 °C for 48 h. MDCK cells were fixed in 4% PFA, followed by ELISA to detect the presence of influenza virus HA proteins. Absorbance readings were converted to percent neutralization (%) using the equation below:% neutralization = ([OD_450_ for negative controls − OD_450_ for sample]/[OD_450_ for negative controls]) × 100%

### 2.12. Statistical Analysis

Graphs for Ab binding and neutralization characteristics were plotted and analyzed using GraphPad Prism (GraphPad, Inc., San Diego, CA, USA). Independent experiments were conducted three times using triplicate wells each time. We did not compare differences between means and, hence, did not determine statistical significance. This is because differences in binding and neutralization profiles for VR-analogous IgG variants were very apparent.

## 3. Results

The root mean square deviation (RMSD) of the C alpha atomic coordinates measures the structural similarity of 3D structures [22]. Zero Angstrom (Å) means 100% similar while RMSD below 2 Å is considered accurate [23]. To ensure predicted IgG structures were accurate, we structurally aligned 3D structures of templates with corresponding predicted models (Figure 1). Predicted structures were considered accurate because RMSD values from structurally aligned 3D structures were all less or equal to 2 Å (Table 2). Minor structural variations were present in some VRs as expected because VR sequences were not identical. Human IgG4 was not predicted due to the absence of an intact IgG4 template in the Protein Data Bank.

Swapping heavy chain constant domains denatures the paratope 3D architecture and influences biochemical characteristics for VR-analogous IgG variants.

Several factors including mutation are known to alter the structural and biochemical characteristics of Ab/proteins [24]. Mutation is defined as any changes to Ab/protein genetic sequences whether large or small [24]. We explored the influence of swapping a human IgG1 heavy chain constant domain with either a mouse IgG1 or mouse IgG2a heavy chain constant domain on the spatial location and orientation of paratope amino acid residues since they directly interact with epitope amino acid residues. To do this, all predicted 3D structures were structurally alignment with the same 3D structure of an HA-bound Fab. Our data show discrepant VR ribbon structures (Figure 2A–C) and spatial locations and orientations of paratope amino acid residues between human IgG1 and mouse IgG1, human IgG1 and mouse IgG2a, and mouse IgG1 and mouse IgG2a (Figure 2D–I). Figure 2B shows predicted mouse IgG1 clashing with HA. Since all model structures were predicted from ligand-unbound templates, the structure assumed by predicted mouse IgG1 might not be the same as that assumed when HA is engaged. To demonstrate this, we structurally aligned 3D structures of a Fab heavy chain not bound to HA- with the heavy chain of the same Fab in complex with HA and exposed paratope amino acid residues including Tyrosine at position 29 (Tyr29). Appendix A shows that the spatial location and orientation of Tyr29 when the Fab is free is very different from that assumed when the Fab engages HA consistent with the definition for induced fit. Induced fit is defined as the repositioning of paratope amino acid residues required to optimize the engagement of ligands leading to maximized Ab specificity and effectiveness [25]. Repositioning and/or rearrangement of paratope amino acid residues was also noted when the human IgG1 was replaced with a mouse IgG1 or mouse IgG2a heavy chain constant domain for VR-analogous CR6261 (Figure 3A–I), FI6 (Figure 4A–I), and 5J8 (Figure 5A–I). These structural discrepancies led to differential interactions with the same HA by the same paratope amino acid residues. Replacing the human IgG1 hinge with a mouse IgG1 hinge also rearranged paratope amino acids residues (Appendix A). To conclude, swapping heavy chain constant domains denatures paratope amino acid residues leading to discordant interactions with HA by VR-analogous IgG variants.

To investigate the impact of heavy chain constant domain-induced denaturation of paratope amino acid residues on Ab binding and neutralization profiles, F045-092, CR6261, FI6, and 5J8 were genetically engineered to possess either a human IgG1/IgG4 or a mouse IgG1/IgG2a heavy chain constant domain. SDS-PAGE data show that recombinant HA and VR-analogous IgG variants were properly expressed and purified (Appendix A). F045-092 prevents the attachment of Influenza viruses to sialic acid receptors by binding to the RBS on HA. However, the human IgG1 does not recognize pH1N1 HA because of a Ser136Thr substitution [26]. Although F045-092 human IgG4 bound NC H1N1, H3N2A, and H3N2P more effectively than human IgG1, they both failed to recognize pH1N1 HA (Figure 2J and Appendix A). Expectedly, F045-092 human IgG1 and IgG4 did not protect MDCK cells from being infected by pH1N1 Influenza viruses (Figure 2M and Appendix A). By contrast, F045-092 with mouse IgG1 and IgG2a heavy chain constant domains bound pH1N1 HA (Figure 2K–L) and prevented pH1N1 Influenza viruses from infecting MDCK cells (Figure 2N–O). By replacing the human IgG1 hinge with a mouse IgG1 hinge, we assessed whether the mouse IgG1 hinge could transfer mouse IgG1-like characteristics to human IgG1. Although human IgG1 with a mouse hinge did not cross-react with pH1N1, binding to NC H1N1, H3N2A, and H3N2P HA was improved (Appendix A). CR6261 binds and neutralizes H1 Influenza viruses [3]. Neither of the VR-analogous CR6261 IgG variants bound H3N2A and H3N2P HA (Figure 3J–L: Appendix A) nor neutralized H3N2 Influenza viruses (Figure 3M–O). Both CR6261 human IgG1 possessing a mouse IgG1 hinge and IgG4 bound H1 HA more effectively than native human IgG1 (Appendix A). Furthermore, we show that VR-analogous CR6261 mouse IgG1, mouse IgG2a, and human IgG1 with a mouse IgG1 hinge protected MDCK cells from being infected by pH1N1 Influenza viruses more effectively since native human IgG1 could not protect 100% of MDCK cells even at the highest concentration (Appendix A). FI6 recognizes and neutralizes group 1 and 2 Influenza viruses [3]. Although FI6 human IgG1 poorly bound H3 HA (Figure 4J) in our hands, VR-analogous human IgG1 with a mouse IgG1 hinge, human IgG4, mouse IgG1, and mouse IgG2a bound H1 and H3 HA more effectively suggesting that the mouse IgG1 hinge introduced mouse IgG1-like characteristics into the native human IgG1 (Figure 4K–L: Appendix A). Although all FI6 VR-analogous IgG variants protected MDCK cells from infection by both pH1NI and H3N2 Influenza viruses, FI6 placed on mouse IgG1 and IgG2a heavy chain constant domains were more effective than human IgG1 (Figure 4M–O). Interestingly, all VR-analogous IgG variants reacted more effectively with H1 than H3 HA or Influenza viruses consistent with current reports by Wu et al. suggesting that H3 Influenza viruses possess a low genetic barrier to resistance compared with H1 Influenza viruses [27]. 5J8 is a pan H1 monoclonal Ab known to bind and neutralize pH1N1 but not NC H1N1 [3]. Our data show that substituting the human IgG1 heavy chain constant domain with either a mouse IgG1 or mouse IgG2a broadens the specificity of 5J8 to include NC H1N1 (Figure 5K–L) which VR-analogous human IgG1 (Figure 5J) and IgG4 (Appendix A) do not recognize. Being a pan-H1 Ab, neither of the VR-analogous IgG variants protected MDCK cells from being infected by H3N2 Influenza viruses (Figure 5M–O). Binding activities for F045-092, CR6261, FI6, and 5J8 human IgG1 against pH1N1 and H3N2 Influenza viruses were further evaluated using Immunofluorescence Assay. Results show that these monoclonal Abs effectively recognized HA within MDCK cells (Appendix A). Taken together, swapping heavy chain constant domains improves and/or broadens the specificity of VR-analogous IgG variants.

F045-092 VR-analogous IgG variants.

**Figure 2 pathogens-12-00051-f002:**
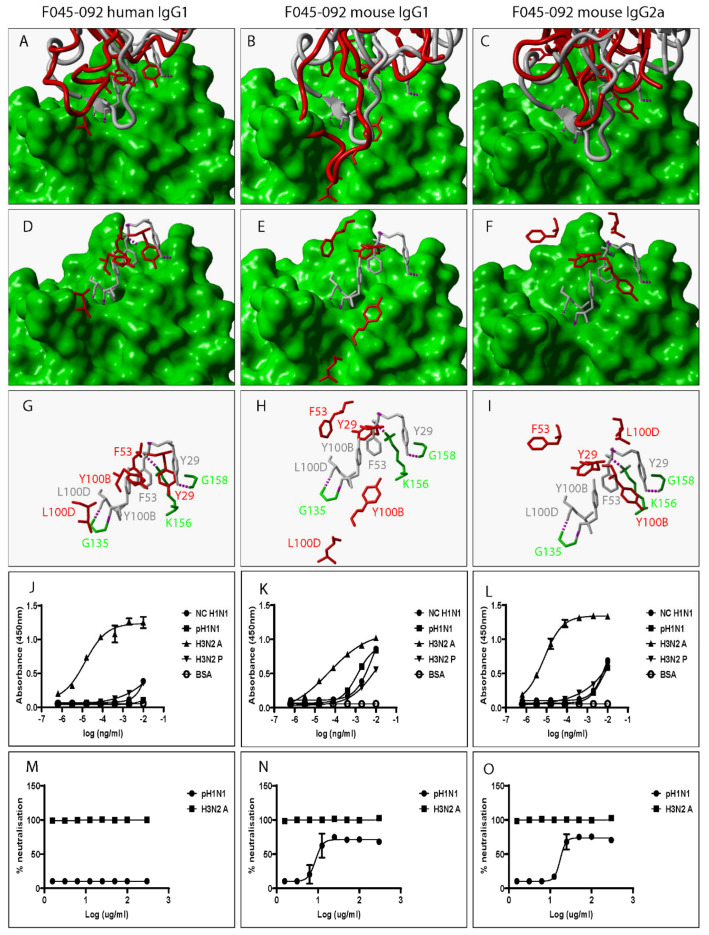
Swapping heavy chain constant domains alters the spatial location and orientation of paratope amino acid residues with specificity and neutralization implications for VR-analogous F045-092 human IgG1, mouse IgG1, and IgG2a. (**A**–**C**): Structural alignment of predicted human IgG1, mouse IgG1, and IgG2a 3D structures (red ribbon) with the same VR-analogous Fab (gray ribbon)—HA1 (green surface) complex (PDB ID—4O58). (**D**–**F**): Exposed paratope amino acid residues of VR-analogous Fab (gray sticks), human IgG1 (red sticks), and mouse IgG1/IgG2a (red sticks). (**G**–**I**): Exposed paratope amino acids (gray sticks) interacting with epitope amino acid residues (green sticks). (**J**–**L**): Binding activity against A/New Caledonia/20/1999 (NC H1N1), A/California/07/2009 (pH1N1), A/Aichi/2/68 (H3N2A), and A/Perth/16/2009 (H3N2P) influenza virus HA proteins. (**M**–**O**): Protection of MDCK cells from pH1N1 and H3N2A influenza virus infection. Hydrogen bonds are presented as dotted magenta lines. Bovine serum albumin (BSA) was used as a negative control. Representative data of four independent experiments performed in triplicate are presented as mean ± SD.

CR6261 VR-analogous IgG variants.

**Figure 3 pathogens-12-00051-f003:**
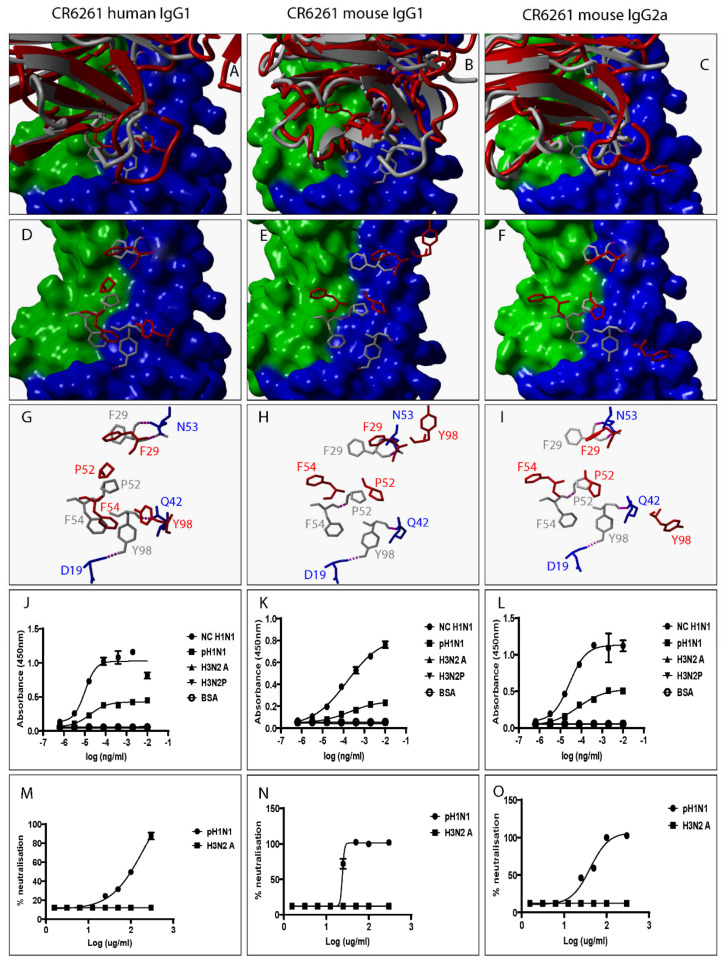
Swapping heavy chain constant domains alters the spatial location and orientation of paratope amino acid residues with specificity and neutralization implications for VR-analogous CR6261 human IgG1, mouse IgG1, and IgG2a. (**A**–**C**): Structural alignment of predicted human IgG1, mouse IgG1, and IgG2a 3D structures (red ribbon) with the same VR-analogous Fab (gray ribbon)-HA2 (blue surface) complex (PDB ID—3GBM). (**D**–**F**): Exposed paratope amino acid residues of VR-analogous Fab (gray sticks), human IgG1 (red sticks), and mouse IgG1/IgG2a (red sticks). (**G**–**I**): Exposed paratope amino acids (gray sticks) interacting with epitope amino acid residues (blue sticks). (**J**–**L**): Binding activity against A/New Caledonia/20/1999 (NC H1N1), A/California/07/2009 (pH1N1), A/Aichi/2/68 (H3N2A), and A/Perth/16/2009 (H3N2P) influenza virus HA proteins. (**M**–**O**): Protection of MDCK cells from pH1N1 and H3N2A influenza virus infection. Hydrogen bonds are presented as dotted magenta lines. Bovine serum albumin (BSA) was used as a negative control. Representative data of four independent experiments performed in triplicate are presented as mean ± SD. Hemagglutinin HA1 (green surface) and HA2 (blue surface).

FI6 VR-analogous IgG variants.

**Figure 4 pathogens-12-00051-f004:**
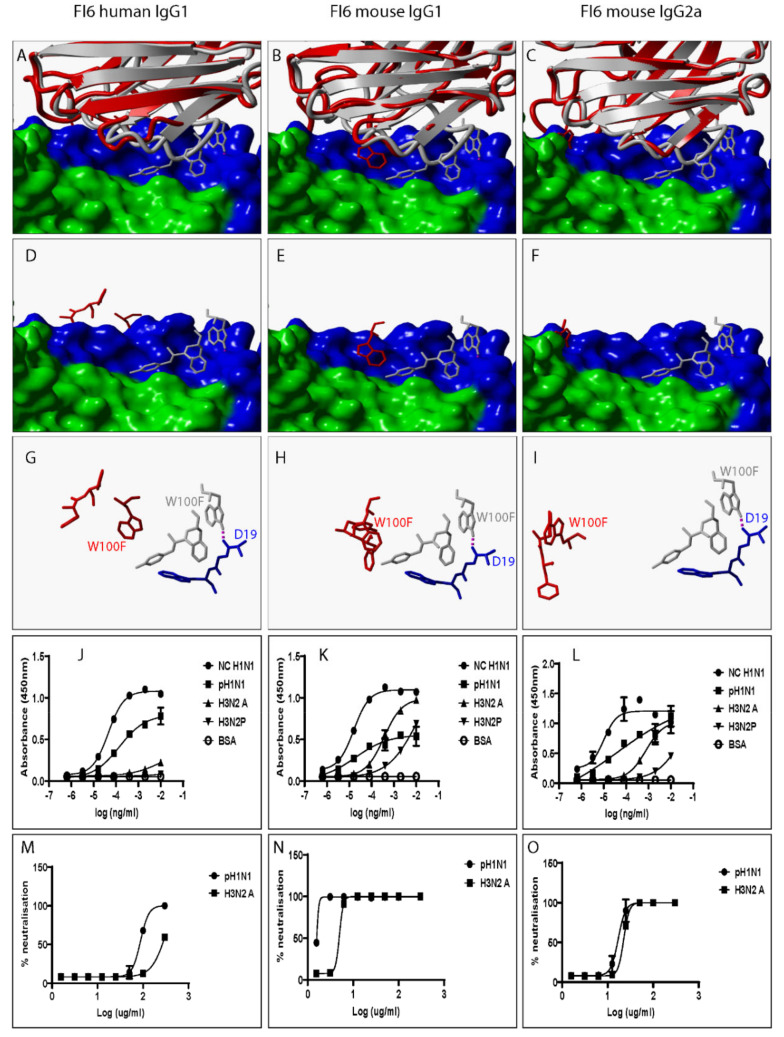
Swapping heavy chain constant domains alters the spatial location and orientation of paratope amino acid residues with specificity and neutralization implications for VR-analogous FI6 human IgG1, mouse IgG1, and IgG2a. (**A**–**C**): Structural alignment of predicted human IgG1, mouse IgG1, and IgG2a 3D structures (red ribbon) with the same VR-analogous Fab (gray ribbon)-HA2 (blue surface) complex (PDB ID—3ZTN). (**D**–**F**): Exposed paratope amino acid residues of VR-analogous Fab (gray sticks), human IgG1 (red sticks), and mouse IgG1/IgG2a (red sticks). (**G**–**I**): Exposed paratope amino acids (gray sticks) interacting with epitope amino acid residues (blue sticks). (**J**–**L**): Binding activity against A/New Caledonia/20/1999 (NC H1N1), A/California/07/2009 (pH1N1), A/Aichi/2/68 (H3N2A), and A/Perth/16/2009 (H3N2P) influenza virus HA proteins. (**M**–**O**): Protection of MDCK cells from pH1N1 and H3N2A influenza virus infection. Hydrogen bonds are presented as dotted magenta lines. Bovine serum albumin (BSA) was used as a negative control. Representative data of four independent experiments performed in triplicate are presented as mean ± SD. Hemagglutinin HA1 (green surface) and HA2 (blue surface).

5J8 VR-analogous IgG variants.

**Figure 5 pathogens-12-00051-f005:**
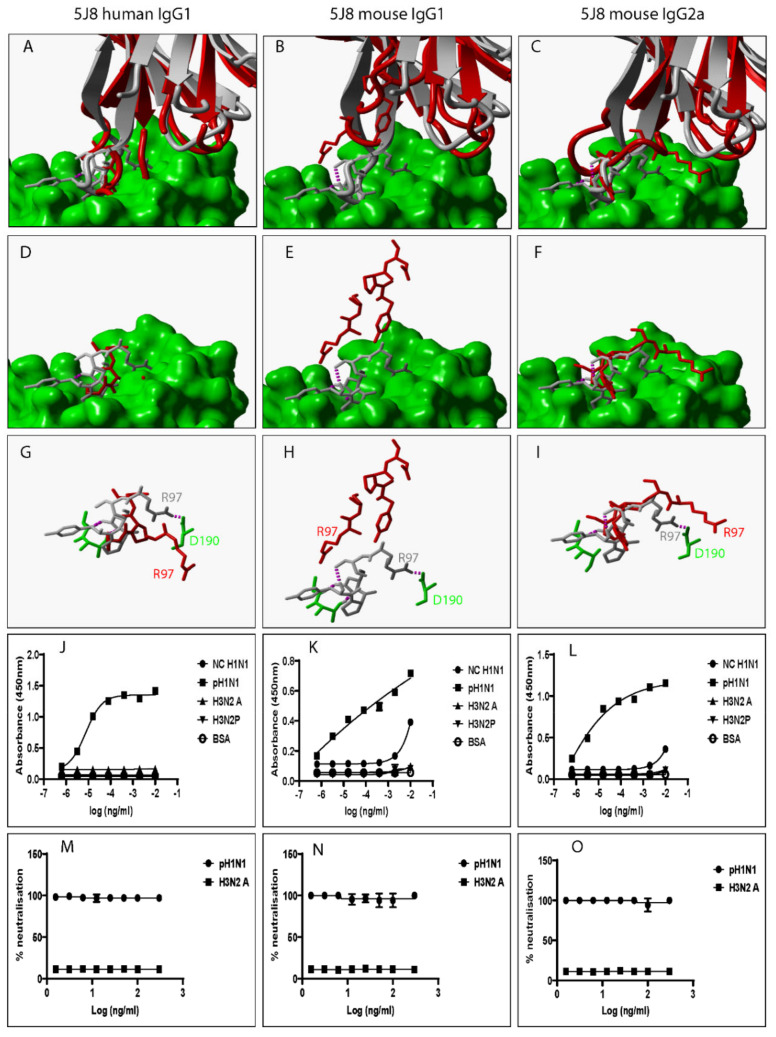
Swapping heavy chain constant domains alters the spatial location and orientation of paratope amino acid residues with specificity and neutralization implications for VR-analogous 5J8 human IgG1, mouse IgG1, and IgG2a. (**A**–**C**): Structural alignment of predicted human IgG1, mouse IgG1, and IgG2a 3D structures (red ribbon) with the same VR-analogous Fab (gray ribbon)-HA1 (green surface) complex (PDB ID—4M5Z). (**D**–**F**): Exposed paratope amino acid residues of VR-analogous Fab (gray sticks), human IgG1 (red sticks), and mouse IgG1/IgG2a (red sticks). (**G**–**I**): Exposed paratope amino acids (gray sticks) interacting with epitope amino acid residues (green sticks). (**J**–**L**): Binding activity against A/New Caledonia/20/1999 (NC H1N1), A/California/07/2009 (pH1N1), A/Aichi/2/68 (H3N2A), and A/Perth/16/2009 (H3N2P) influenza virus HA proteins. (**M**–**O**): Protection of MDCK cells from pH1N1 and H3N2A influenza virus infection. Hydrogen bonds are presented as dotted magenta lines. Bovine serum albumin (BSA) was used as a negative control. Representative data of four independent experiments performed in triplicate are presented as mean ± SD.

Hydrogen bonds are critical for binding and complex stabilization. For hydrogen bonds to form, the distance between donor and acceptor atoms should be less than 5 Å [13]. To explore whether paratope amino acid residues in VR-analogous IgG variants will form hydrogen bonds with the same complementary epitope amino acid residues as seen in 3D structures of Fab-HA complexes, we used YASARA View to measure distances between the same donor/acceptor atoms on paratope amino acid residues for HA-bound Fabs or free Fabs or free human IgG1 or free mouse IgG1 or free mouse IgG2a and the same acceptor/donor atoms on epitope amino acid residues (Table 3). The data show variations in the orientation and sometimes the location of paratope amino acid residues between VR-analogous HA-bound and free Fabs. Although the distance between paratope donor/acceptor atoms and epitope acceptor/donor atoms in free Fabs was mostly less than 5 Å, indicating the potential for forming hydrogen bonds, exceptions were noted in which the distance was greater than 5 (Å) and hydrogen bonds were established, nonetheless. Examples include interactions between Tyr29 and Gly158 (8.5 Å), and between Tyr98 and Asp19 (6.2 Å) in F045-092 and CR6261 paratopes, respectively indicative of induced fit (Table 3). Based on the principle of induced fit, it would not be unreasonable to suggest that paratope and epitope amino acid residues with distances greater than 5 Å but less than or equal to 8.5 Å between them can potentially form hydrogen bonds. These are illustrated with asterisks in Table 3. Most distances between paratope and epitope amino acid residues by far exceeded 8.5 Å implying that induced fit might not be a possibility. These exceedingly repositioned and/or reoriented paratope amino acid residues most likely interact with different epitope amino acid residues which might explain discrepancies between VR-analogous Fabs and their VR-analogous parent IgG or between IgG variants. FI6 data is missing in Table 3 because no 3D structure for a free FI6 Fab was found in the Protein Data Bank.

## 4. Discussion

Influenza viruses pose a very serious threat to both public health and the economy and measures to prevent and control their acquisition and spread are urgently needed. Monoclonal Abs are one of the measures. To date, there are no FDA-approved therapeutic or prophylactic monoclonal Abs for Influenza. This is because human monoclonal Abs evaluated in clinical trials failed to show a health benefit relative to comparators despite demonstrating superiority in animal models including mice [4]. Human monoclonal Abs are genetically engineered to possess mouse IgG heavy chain constant domains prior to being evaluated for protective efficacy in mouse models [8,9,10]. We hypothesized that swapping human IgG heavy chain constant domains with mouse IgG heavy chain constant domains will alter the structural and, by extension, Ab specificity and effectiveness.

Ab binding and neutralization depend on 3D structures which also depend on amino acid sequences [24]. Abs possess two identical light and heavy polypeptide chains. The heavy chain is a single polypeptide containing sequences for the variable and the Fc domain containing heavy chain constant domains. Ab Fc-domains can fold autonomously, are interchangeable, and mediate immune effector functions. Similarly, Ab Fab-domains can fold autonomously, encode variable regions (VRs), and mediate Ab specificity. Any genetic alteration of the heavy chain sequence will constitute a mutation which should remain true whether mutations are introduced in variable or heavy chain constant domain encoding sequences and should play an active role in denaturing 3D structures of Abs because mutation, just like chemical denaturants, pH, force, pressure, and temperature can denature 3D structures of Abs which in turn alters Ab specificity and effectiveness [24]. The influence of swapping heavy chain constant domain sequences on the denaturation of paratope amino acid residues is highly disputed because heavy chain constant domains do not directly interact with antigens and their core function is confined to inducing Fc receptor-mediated immune effector functions since they contain the Fc domain. Here, we employ homology modeling to predict 3D structures for VR-analogous human IgG1/IgG4 and mouse IgG1/IgG2a monoclonal Abs. VR-analogous 3D structures for F045-092, CR6261, FI6, and 5J8 monoclonal Abs were used to investigate the role/contribution of heavy chain constant domains in determining/optimizing the paratope 3D fine architecture. In addition, we genetically engineered, expressed, and purified VR-analogous IgG monoclonal Ab variants and used these to explore the influence of swapping heavy chain constant domains on binding and neutralization profiles. The fact that the number of antigen-binding sites (paratopes), light chains, VR sequences, and HA (recombinant or on Influenza viruses) are all kept constant gives us the confidence to explain any inconsistencies between VR-analogous IgG variants to be a consequence of variations in their heavy chain constant domains. We found that mutating entire heavy chain constant domains altered the 3D fine architecture of paratope amino acid residues suggesting that VR-analogous IgG variants might not share the same paratope 3D fine architecture and might not be interacting with the same epitope amino acid residues on HA. This most likely explains discordant binding and neutralization characteristics exhibited by VR-analogous monoclonal Abs. This observation was consistent for F045-092, CR6261, FI6, and 5J8 monoclonal Abs indicating that heavy chain constant domain-induced denaturation of paratope 3D fine architectures is generalizable and that this property is neither HA epitope-dependent (RBD versus fusion peptide) nor Ab breadth-dependent (binds to pan-H1 [5J8], group 1 [CR6261], or group 1 and 2 [FI6 and F045-092] HA). We noted that CR6261 did not manifest broadened specificity and neutralization to H3N2 HA/Influenza viruses an indication that heavy chain constant domain-induced conformational changes of paratope amino acid residues alone are insufficient for broadening Ab specificity and neutralization characteristics. Other factors such as the Influenza virus strain used could play a role. However, it should be noted that only a small panel of Influenza HA and viruses was used implying that results could be different if a large panel of Influenza viruses is used. Binding and neutralization characteristics of VR-analogous IgG1, IgG4, and IgG1 with a mouse IgG1 hinge improved but did not broaden Ab specificity. By contrast, VR-analogous mouse IgG1 and mouse IgG2a did improve and broaden Ab specificity. This can be explained by the fact that paratope 3D architectures of VR-analogous human IgG variants are mildly denatured due to high sequence similarity between them whereas those for VR-analogous mouse IgG1 and mouse IgG2a are extensively denatured due to reduced similarities between human IgG1 and mouse IgG1/IgG2a heavy chain constant domains. This is consistent with reports suggesting that similarities between 3D structures of Abs/proteins decrease as sequence variation increases [15,18].

By aligning 3D structures of B12 Fab and B12 IgG1, we showed that deleting the Fc domain sequence reorients paratope amino acid residues and alters not only binding and neutralization characteristics but the mechanisms of protection as well [24]. F045-092 and 5J8 have been reported to exhibit discordant binding profiles against the same strains of Influenza viruses [26,28]. The explanation given by several studies is that the inconsistency is due to a Fab having one paratope whereas the VR-analogous IgG1 has two paratopes suggesting that bivalency advantages the parent IgG1 over its Fab. However, by comparing a bivalent F(ab’)2 with a bivalent VR-analogous IgG1 and still finding functional discrepancies, a study by Suryadevara et al. comprehensively demonstrated that Ab bivalency does not explain the IgG1/Fab of IgG1/ F(ab’)2 discordance [29]. This study still did not provide an explanation for the discrepancy despite being comprehensive. Our data fully support their conclusion and further suggest that the inconsistency might be explained by mild structural denaturation of paratope amino acid residues driven by genetically or chemically deleting the Fc domain sequence consistent with our previous findings [24]. Decreased protection of MDCK cells from infection by Influenza viruses by a mouse monoclonal Ab (6F12) placed on a human IgG1 heavy chain constant domain compared with the native VR-analogous mouse IgG1 and IgG2a was reported [8]. These data imply that VR-analogous IgG variants placed on mouse heavy chain constant domains are superior to those placed on human IgG heavy chain constant domains regardless of the species of origin. Although we did not explore the in vivo protective efficacy of VR-analogous IgG variants in mouse models, some studies have reported broadened protective efficacy in vivo. However, the broadened protective efficacy was explained by Fc receptor-mediated immune effector functions [10]. Although Fc receptor-mediated immune effector functions contribute to viral clearance, similar studies have concluded that Fc receptor-mediated immune effector functions become dispensable as the concentration of therapeutic/prophylactic Abs is increased [8]. Another similar study concluded that Fc receptor-mediated immune effector functions did not explain why some of the Abs (non-neutralizing) they investigated exhibited broadened protective efficacy in mouse models [9]. Our data suggest that in vivo protective efficacy in mouse models using VR-analogous human monoclonal Abs placed on mouse heavy chain constant domains could be explained by mutation-induced structural variations in paratope amino acid residues.

Our findings have implications for studies using mouse models to assess the therapeutic or prophylactic efficacy of human monoclonal Abs formatted to possess mouse IgG heavy chain constant domains. The concept behind swapping a human with a mouse heavy chain constant domain is to ensure compatibility between mouse Fc domains and Fc receptors on mouse immune effector cells. We have shown that VR-analogous human IgG1 and mouse IgG variants do not possess the same/similar structural and biochemical characteristics implying that the two Ab variants are not the same. Therefore, inferences from VR-analogous mouse IgG variants to human IgG1 should be made with extreme caution. Some studies directly use human monoclonal Abs in mouse models. Findings from such study designs should also be interpreted with extreme caution since the interaction between the human Fc domain and mouse Fc receptors might not be compatible/optimal. Our data supported by others suggest that Abs induced by the same Influenza vaccine candidate will most likely demonstrate greater protective efficacy in mouse models relative to human subjects with a caveat that elicited Abs in human subjects and mouse models possess analogous VR. We also suggest that Abs elicited in mouse models will most likely exhibit broadened specificity. This might be because of variations in the quality and not the quantity of vaccine-elicited Abs. We recommend that the protective efficacy of human monoclonal Abs and Influenza vaccine candidates should be evaluated in humanized rather than traditional mouse models. Although this comes with an extra budget, the price for proceeding to clinical trials and failing because of biased preclinical study designs and/or flawed inferences drawn from them is not cheap either. Alternatively, traditional mouse models can be used to explore initial protective efficacy studies followed by validation in humanized mouse models if successful. Whether this is applicable to other animal models should be explored.

In conclusion, swapping heavy chain constant domains alters the structural and biochemical characteristics of VR-analogous IgG variants. These data have implications for assessing the protective efficacy of monoclonal Abs or Influenza vaccine candidates in traditional mouse models.

Our study has some limitations. First, we did not use mouse Abs to explore whether swapping the heavy chain constant domain for an original mouse Ab with a human heavy chain constant domain will have a similar effect on the paratope 3D architecture. However, this was demonstrated by others as illustrated above [8]. Second, since actual crystal structures were not employed in our study, we urge a cautious interpretation of our findings as some observations could be artifacts. However, our data are supported by actual Fab and IgG1 3D structures for B12 Ab [24]. We also did not predict potential epitopes for denatured paratope amino acid residues. However, there is a possibility that predictions might be inaccurate due to induced fit. Both limitations underscore the need for utilizing actual crystal structures in future studies. Importantly, our study serves as a strong foundation/justification for conducting comprehensive studies in the future.

## Figures and Tables

**Figure 1 pathogens-12-00051-f001:**
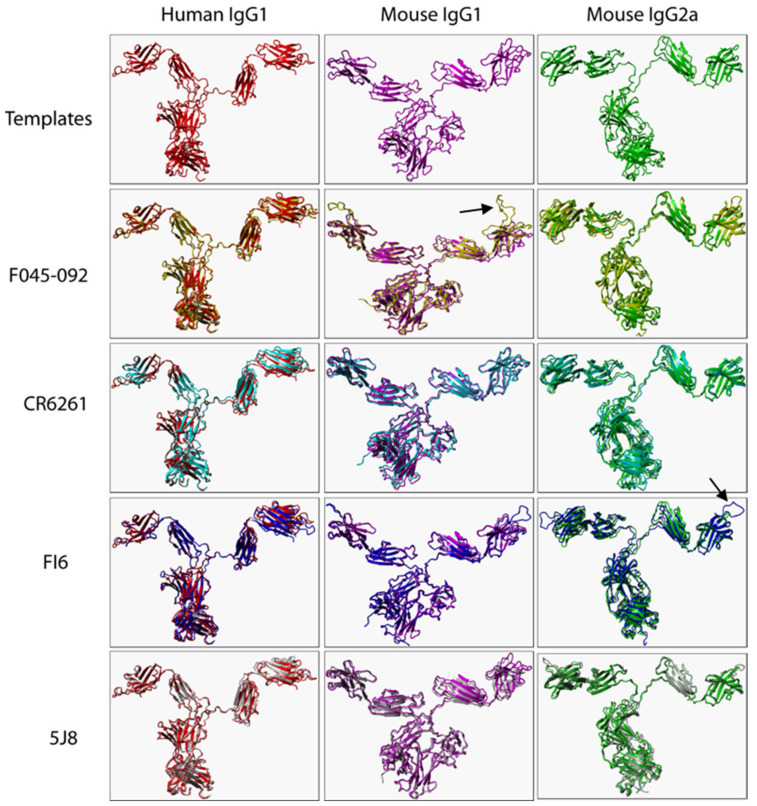
Structural alignment of B12 (human IgG1), Mab-61.1.3 (mouse IgG1), and MAB231 (mouse IgG2a) (templates) with the corresponding predicted 3D structures of F045-092, CR6261, FI6, and 5J8. Only heavy chains (VH, hinge region, CH1-CH3) of templates and predicted mouse models are presented. Arrows illustrate structural variations in VRs.

**Table 1 pathogens-12-00051-t001:** Characteristics of 3D crystal structures included in this study.

PDB ID (Ligand Status)	Ab Name	Ligand	Ab Formulation	Method
1HZH (unbound)	B12	HIV1-gp120	Human IgG1	X-ray diffraction
1IGY (unbound)	Mab-61.1.3.	Phenobarbital	Mouse IgG1	X-ray diffraction
1IGT (unbound)	MAB231	Canine lymphoma	Mouse IgG2a	X-ray diffraction
4O5L (unbound)	F045-092	Influenza HA (H1, H2, H3, and H5)	Fab	X-ray diffraction
4EVN (unbound)	CR6261	Influenza HA (H1, H2, H5, H6, H8, H9, H11, H12, H13, H16)	Fab	X-ray diffraction
4M5Y (unbound)	5J8	Influenza HA (Pan-H1)	Fab	X-ray diffraction
4O58 (HA-bound)	F045-092	Influenza HA (H1, H2, H3, and H5)	Fab	X-ray diffraction
3GBM (HA-bound)	CR6261	Influenza HA (H1, H2, H5, H6, H8, H9, H11, H12, H13, H16)	Fab	X-ray diffraction
3ZTN (HA-bound)	FI6	Influenza HA (H1, H2, H3, H4, H5, H6, H7, H8, H9, H10, H11, H12, H13, H14, H15, and H16)	Fab	X-ray diffraction
4M5Z (HA-bound)	5J8	Influenza HA (Pan-H1)	Fab	X-ray diffraction

**Table 2 pathogens-12-00051-t002:** RMSD values (Å) of aligned X-ray templates with corresponding predicted 3D structures.

Intact IgG Templates (PDB ID)	5J8	F045-092	CR6261	FI6
Human IgG1 (1HZH)	0.545	0.568	0.403	0.484
Mouse IgG1 (1IGY)	1.038	1.473	1.564	0.707
Mouse IgG2a (1IGT)	1.334	0.868	1.869	2.011

**Table 3 pathogens-12-00051-t003:** Distances between donor/acceptor atoms on paratope amino acid residues and acceptor/donor atoms epitope amino acid residues of VR-analogous human and mouse IgG variants.

VR Identical IgG (Class)	HCDR	Interacting Amino Acid Residues		Distance between Interacting Amino Acid Residues (Å)	Potential for Forming Hydrogen Bonds
		Paratope	Epitope		
F045-092 (Free human IgG1)	1	Tyr 29	Gly 158	3.67	Yes
	2	Phe 53	Lys 156	2.82	Yes
	3	Tyr 100B	Gly 135	10.45	No
		Leu 100D	Gly 135	11.68	No
F045-092 (Free mouse IgG1)	1	Tyr 29	Gly 158	12.64	No
	2	Phe 53	Lys 156	7.08 *	No
	3	Tyr 100B	Gly 135	10.39	No
		Leu 100D	Gly 135	11.37	No
F045-092 (Free mouse IgG2a)	1	Tyr 29	Gly 158	14.85	No
	2	Phe 53	Lys 156	11.29	No
	3	Tyr 100B	Gly 135	13.93	No
		Leu 100D	Gly 135	17.3	No
F045-092 (Free Fab)	1	Tyr 29	Gly 158	8.5 *	No
	2	Phe 53	Lys 156	4.3	Yes
	3	Tyr 100B	Gly 135	2.7	Yes
		Leu 100D	Gly 135	3.1	Yes
F045-092 (Fab-HA bound)	1	Tyr 29	Gly 158	2.8	Yes
	2	Phe 53	Lys 156	3.1	Yes
	3	Tyr 100B	Gly 135	2.7	Yes
		Leu 100D	Gly 135	2.9	Yes
CR6261 (Free human IgG1)	1	Phe 29	Asp 53	6.18 *	No
		Phe 29	Asp 53	7.54 *	No
	3	Tyr 98	Asp 19	10.13	No
		Tyr 98	Gln 42	10.64	No
CR6261 (Free mouse IgG1)	1	Phe 29	Asp 53	7.22 *	No
		Phe 29	Asp 53	7.21 *	No
	3	Tyr 98	Asp 19	32.78	No
		Tyr 98	Gln 42	20.49	No
CR6261 (Free mouse IgG2a)	1	Phe 29	Asp 53	7.29 *	No
		Phe 29	Asp 53	7.37 *	No
	3	Tyr 98	Asp 19	17.7	No
		Tyr 98	Gln 42	12.09	No
CR6261 (Free Fab)	1	Phe 29	Asp 53	3.8	Yes
		Phe 29	Asp 53	3.5	Yes
	3	Tyr 98	Asp 19	6.2 *	No
		Tyr 98	Gln 42	3.3	Yes
CR6261 (Fab-HA bound)	1	Phe 29	Asp 53	2.9	Yes
		Phe 29	Asp 53	2.9	Yes
	3	Tyr 98	Asp 19	2.8	Yes
		Tyr 98	Gln 42	2.8	Yes
5J8 (Free human IgG1)	3	Arg 97	Asp 190	7.91 *	No
		Asp 100B	Thr 137	8.16 *	No
5J8 (Free mouse IgG1)	3	Arg 97	Asp 190	11	No
		Asp 100B	Thr 137	31.53	No
5J8 (Free mouse IgG2a)	3	Arg 97	Asp 190	10.78	No
		Asp 100B	Thr 137	5.87	No
5J8 (Free Fab)	3	Arg 97	Asp 190	2.6	Yes
		Asp 100B	Thr 137	1.6	Yes
5J8 (Fab-HA bound)	3	Arg 97	Asp 190	2.2	Yes
		Asp 100B	Thr 137	2.8	Yes

HCDR—heavy-chain complementarity determining region. Asterisk (*)—paratope amino acid residues have potential to form hydrogen bonds by induced fit.

## Data Availability

Not applicable.

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
