# Peer review of "Switching Heavy Chain Constant Domains Denatures the Paratope 3D Architecture of Influenza Monoclonal Antibodies"

_pathogens, 2022, doi:10.3390/pathogens12010051_

Round 1

Reviewer 1 Report

This article describes a methodology to study the interactions between Fc fragments of immunoglobulins and the paratope of an antibody, precisely the impact of Fc domain swapping on the paratope structure and function. To do this, the authors produce chimeric antibodies that combine human parts (containing the variable domains) with fragments of murine origin, and evaluate their ability to bind to the antigen as well as the neutralizing capacity, compared to the native antibody. Even though the paper is well written, I have some questions about the rational of the study.

As the objective is to construct some IgGs with swapped Fc-domains, why not using only human sequences of Fc variants? The justification for the use of mouse Fc needs to be better argued. Indeed Fc fragment swapping can occur during natural immunoglobulin maturation and production process, and in this case I hardly understand how the chimeric human/murine model presented is illustrative of the molecular arrangements that take place in reality. On the other hand, if the goal is to illustrate the swapping that could take regarding the production process of therapeutic antibodies, usually the Fc fragments of recombinant antibodies are from human origin, so I have the same concern, because chimeric therapeutic antibodies are note those described in this study. Thank you for clarifying carefully this point.

Here are thereafter some suggestions:

Introduction:

- Authors talk about the VR regions regarding the Fc regions interactions. As an information, perhaps they can also talk about the CDR and framework regions (FR) interactions, as punctual mutations in FR regions also influence paratope conformation and epitope recognition.

- L52: Incomplete sentence?

“…inconsistent with reports that showed that Abs with swapped Fc domains…

M&M section:

- Ab variants: authors should present a summary table of the characteristics of these model Ab and better justify this choice, otherwise the reader is obliged to look for the references

- Expression of HA and Ab variants: authors don’t show any result about the efficacy of the purification processes. Are HA ant Abs sufficiently purified so they can carry on the binding experiments? What are their purity criteria?

- ELISA: please specify in the texts the number of independent experiments (triplicates).

- Statistical analysis: please state the statistical tests that are used here.

Results Section:

L202: I do not agree to say that F045-092 huma IgG1 binds H3N2A and H3N2P, as the binding is very weak and only for high antibody concentrations.

Discussion:

- L336-338: The sentence “Additionally…viral escape mutants” is difficult to understand. Could you please better explain using multiple sentences instead of this long one?

Author Response

Thank you for your time, the positive feedback, and the desire to improve our manuscript. Please see our responses below. Thank you.

Introduction:

- Authors talk about the VR regions regarding the Fc regions interactions. As an information, perhaps they can also talk about the CDR and framework regions (FR) interactions, as punctual mutations in FR regions also influence paratope conformation and epitope recognition.

Our response

Included. Please see our revised manuscript.

- L52: Incomplete sentence?

Our response

Please see our revised manuscript.

“…inconsistent with reports that showed that Abs with swapped Fc domains…

 Our response

Please see our revised manuscript.

M&M section:

- Ab variants: authors should present a summary table of the characteristics of these model Ab and better justify this choice, otherwise the reader is obliged to look for the references

Our response

Included. Please see our revised manuscript.

- Expression of HA and Ab variants: authors don’t show any result about the efficacy of the purification processes. Are HA ant Abs sufficiently purified so they can carry on the binding experiments? What are their purity criteria?

Our response

Included. Please see our revised manuscript.

- ELISA: please specify in the texts the number of independent experiments (triplicates).

- Statistical analysis: please state the statistical tests that are used here.

Our response

Included. Please see our revised manuscript.

Results Section:

L202: I do not agree to say that F045-092 huma IgG1 binds H3N2A and H3N2P, as the binding is very weak and only for high antibody concentrations.

Our response

We have included IFA results to show that F045-092 interacts with H3N2A. Please see our revised manuscript.

Discussion:

- L336-338: The sentence “Additionally…viral escape mutants” is difficult to understand. Could you please better explain using multiple sentences instead of this long one?

Our response

Please see our revised manuscript.

Reviewer 2 Report

Review of Moffat. et.al.

The manuscript by Moffat et.al., utilizes homology modeling to predict that the conformation of paratopes of antibodies targeting influenza hemagglutinin (HA) changes upon switching the class or species of the constant region of the antibody. In addition, they present experimental data showing that the specificity of binding to subtypes of influenza HA can be altered by swapping the hinge region of human antibodies with that of mouse antibodies.

This study is based on previous findings that switching the isotype or using chimeric constant regions, alters the efficacy of neutralizing activity and specificity of anti-viral antibodies. The study design shows data from two independent projects as the constructs used for the homology modeling and experimental validation are different (mouse constant region vs mouse hinge region). The manuscript is difficult to read and is missing some experimental details needed.

Below are my comments that need to be addressed.

1.       The authors state in the abstract (lines 22-25), “We used VR analogous Abs to demonstrate that swap-22 ping Fc-domains can directly modulate the fine structure of paratopes in tethered Fab-domains, 23 resulting in the exhibition of discordant binding and neutralization characteristics that are distinct 24 from those linked to simple alterations in intra-Ab Fab-Fab or Fab-Fc distances.”

However, in the manuscript the work describes exchanging the whole constant region including CH1 domain (which is not part of Fc) or the hinge region. Using the term “swapping Fc domains” loosely is misleading and inappropriate. The authors need to rephrase the abstract to reflect this. There are several instances in discussion which need to be updated as well where Fc domain swapping is used.

2.       In lines 56-59, the authors state the homology modeling can be accurate when there is greater than 50% identity between template and comparator sequences. What is the sequence identity of paratopes of F045-092, CR6261, FI6, 5J8 with the paratope of the template? Is there sufficient homology to predict the paratope conformations?  While the global homology can be used to predict secondary structure in folded regions, can the same models be used to predict CDR conformations where the sequences diverge from the template?

3.       In lines 212-217, the authors state “Predicted human and human-mouse IgG structural alignment with Fab-HA complex crystal structures revealed VR conformational differences for Fab, human, and human-mouse Ab variants (Fig. 2G, H, and I). Further examination showed that amino acids in the paratope were rearranged leading to perturbed paratope-epitope interactions which explains inconsistent binding and neutralization characteristics for F045-092 Ab variants (Fig. S3).”

In Figure 2G, H and I, the authors align the human IgG1, mouse IgG1 and mouse IgG2a versions of predictions with PDB 4O58. Based on the predictions, in Figure 2H, the paratope of F045-092, seems to sterically clash with H3N2 HA. However, the antibody seems to be effective in binding and neutralizing H3N32P virus. Does this mean that the predicted paratope and subsequent inferences on differential binding are inaccurate? The authors make similar observations in Figures 3, 4 and 5 regarding differences in paratope conformations. I am not sure if the modeling data is conclusive and hence the authors need to be careful in claiming that paratopes are significantly altered.

4.       There are several aspects of the discussion that are difficult to read and need more elaboration. This is difficult for a non-expert in the field to follow.

For example, Lines 313-314, there is a discussion of “avidity hypothesis”, but nowhere in the text is this hypothesis described.

In lines 310-311, the authors state “This proposed mechanism does not explain the observed functional loss of activity against one influenza strain but not others by F045-092 and 5J8 Abs”. This needs more explanation. A single paratope can explain differences in binding as there can be differences in HA epitope across subtypes.

I would suggest editing/rewriting the discussion to make it more readable.

5.       Following missing experimental details need to be addressed:

i)                    Expression vector details and amino acid sequences of the expressed antibody constructs need to be mentioned in either main text or supplemental information.

ii)                   Provide reference for plasmids used from other studies (like J-chain)

iii)                 In lines 128-129, the authors say equal amounts but do not specify the amount of antibody used in neutralization assay.

Overall, the manuscript is interesting and the experimental findings with hinge substituted human mouse chimeras are novel. It is interesting to note that hinge and CH1 domain interactions are important in flexibility of Fab arms (Schneider et.al, https://www.pnas.org/doi/abs/10.1073/pnas.85.8.2509) and mispairing of hinge CH1 domains may have effects on paratope.

Author Response

Thank you for your time, the positive feedback, and the desire to improve our manuscript. Please see our responses below. Thank you.

Below are my comments that need to be addressed.

  1. The authors state in the abstract (lines 22-25), “We used VR analogous Abs to demonstrate that swap-22 ping Fc-domains can directly modulate the fine structure of paratopes in tethered Fab-domains, 23 resulting in the exhibition of discordant binding and neutralization characteristics that are distinct 24 from those linked to simple alterations in intra-Ab Fab-Fab or Fab-Fc distances.”

However, in the manuscript the work describes exchanging the whole constant region including CH1 domain (which is not part of Fc) or the hinge region. Using the term “swapping Fc domains” loosely is misleading and inappropriate. The authors need to rephrase the abstract to reflect this. There are several instances in discussion which need to be updated as well where Fc domain swapping is used.

Our response

We have revised the manuscript. Please see our revised manuscript.

  1. In lines 56-59, the authors state the homology modeling can be accurate when there is greater than 50% identity between template and comparator sequences. What is the sequence identity of paratopes of F045-092, CR6261, FI6, 5J8 with the paratope of the template? Is there sufficient homology to predict the paratope conformations?  While the global homology can be used to predict secondary structure in folded regions, can the same models be used to predict CDR conformations where the sequences diverge from the template?

Our response

Yes. Although the similarity exceeds the 50% threshold, we are not sure if this can be determined in small units as they may not be folding autonomously. We have revised the manuscript to address your valid concern. Please see our revised manuscript.

  1. In lines 212-217, the authors state “Predicted human and human-mouse IgG structural alignment with Fab-HA complex crystal structures revealed VR conformational differences for Fab, human, and human-mouse Ab variants (Fig. 2G, H, and I). Further examination showed that amino acids in the paratope were rearranged leading to perturbed paratope-epitope interactions which explains inconsistent binding and neutralization characteristics for F045-092 Ab variants (Fig. S3).”

In Figure 2G, H and I, the authors align the human IgG1, mouse IgG1 and mouse IgG2a versions of predictions with PDB 4O58. Based on the predictions, in Figure 2H, the paratope of F045-092, seems to sterically clash with H3N2 HA. However, the antibody seems to be effective in binding and neutralizing H3N32P virus. Does this mean that the predicted paratope and subsequent inferences on differential binding are inaccurate? The authors make similar observations in Figures 3, 4 and 5 regarding differences in paratope conformations. I am not sure if the modeling data is conclusive and hence the authors need to be careful in claiming that paratopes are significantly altered.

Our response

Model IgG 3D structures were predicted from templates that did not engage antigens. Therefore, one explanation could be that the free position could not be the same position assumed following Antigen engagement. We have revised the manuscript to address your valid concern. Please see our revised manuscript.

  1. There are several aspects of the discussion that are difficult to read and need more elaboration. This is difficult for a non-expert in the field to follow.

For example, Lines 313-314, there is a discussion of “avidity hypothesis”, but nowhere in the text is this hypothesis described.

In lines 310-311, the authors state “This proposed mechanism does not explain the observed functional loss of activity against one influenza strain but not others by F045-092 and 5J8 Abs”. This needs more explanation. A single paratope can explain differences in binding as there can be differences in HA epitope across subtypes.

I would suggest editing/rewriting the discussion to make it more readable.

Our response

We have revised the manuscript to address your valid concern. Please see our revised manuscript.

  1. Following missing experimental details need to be addressed:
  2. i)Expression vector details and amino acid sequences of the expressed antibody constructs need to be mentioned in either main text or supplemental information.
  3. ii)Provide reference for plasmids used from other studies (like J-chain)

iii)                 In lines 128-129, the authors say equal amounts but do not specify the amount of antibody used in neutralization assay.

Our response

We have revised the manuscript to address your valid concerns. Please see our revised manuscript.

Round 2

Reviewer 1 Report

This new version of the article is presented in a more comprehensible and better argued way. The authors have responded to the questions and requests that have been made, and have supplemented the manuscript with additional figures.

I give a favorable opinion to the publication